# Structural Studies of Henipavirus Glycoproteins

**DOI:** 10.3390/v16020195

**Published:** 2024-01-27

**Authors:** Aaron J. May, Priyamvada Acharya

**Affiliations:** 1Duke Human Vaccine Institute, Duke University, Durham, NC 27710, USA; 2Department of Biochemistry, Duke University, Durham, NC 27710, USA; 3Department of Surgery, Duke University, Durham, NC 27710, USA

**Keywords:** Henipavirus, RNA virus, glycoprotein, viral entry

## Abstract

Henipaviruses are a genus of emerging pathogens that includes the highly virulent Nipah and Hendra viruses that cause reoccurring outbreaks of disease. Henipaviruses rely on two surface glycoproteins, known as the attachment and fusion proteins, to facilitate entry into host cells. As new and divergent members of the genus have been discovered and structurally characterized, key differences and similarities have been noted. This review surveys the available structural information on *Henipavirus* glycoproteins, complementing this with information from related biophysical and structural studies of the broader *Paramyxoviridae* family of which Henipaviruses are members. The process of viral entry is a primary focus for vaccine and drug development, and this review aims to identify critical knowledge gaps in our understanding of the mechanisms that drive *Henipavirus* fusion.

## 1. Introduction

### 1.1. Henipavirus Glycoproteins

Henipaviruses are enveloped, negative-sense, and non-segmented RNA viruses of the *Paramyxoviridae* family that must engage in membrane fusion to enter host cells. Two surface glycoproteins, the attachment and fusion proteins, together facilitate the fusion between virus and host cell membranes. As the sole surface proteins on *Henipavirus* virions, they are also the primary target for neutralizing antibodies, making their antigenicity and functionality topics that are highly relevant for the development of vaccines and therapies.

The attachment protein, referred to as “G” in Henipaviruses, is a single-pass type II transmembrane protein with an N-terminal cytoplasmic domain, a long alpha-helical stalk, and a beta propeller domain known as the head or receptor-binding domain (Figure 1A). The oligomerization of attachment proteins is driven by the stalk domain, where disulfide bonding allows for the formation of a tetramer (dimer of dimers) [1,2,3]. Although recent structural studies that will be addressed in this review establish the tetrameric structure of Henipavirus G proteins [3,4], it remains unclear if the tetrameric state is the only biologically relevant state as one tomographic reconstruction of a related attachment protein from a non-Henipavirus *Paramyxoviridae* member was more consistent with the presence of a dimer rather than a tetramer [5].

The fusion protein, known as F, is also a single-pass transmembrane protein with the cytosolic domain located at the C-terminal end of the protein. As a class I fusion protein, F contains domains critical for facilitating fusion, including a hydrophobic fusion peptide and heptad repeat domains [7]. Many types of class I fusion proteins are utilized across diverse viral families; consequently, the specific sizes, shapes, and mechanisms of these proteins may differ. There are certain conserved features of class I fusion proteins, however, that are essential to the fusion promotion process and are shared by any class I fusion protein, those of Henipaviruses included. These features include the existence of a metastable pre-fusion conformation, use of a hydrophobic fusion peptide, and conversion to the post-fusion state controlled by a triggering signal (Figure 1B). The fusion peptide embeds in host membranes, anchoring the virus to the cell and allowing membrane fusion to proceed. These features are reviewed in detail by White et al. [8]. A feature that distinguishes *Henipavirus* F proteins from other class I examples is the separation of cell attachment or receptor binding and fusion roles into separate proteins. Therefore, the conformational steps that allow receptor binding to lead to the class I fusion promotion steps that are outlined by White et al. are likely quite different from other viruses even if the general steps driving fusion remain similar.

### 1.2. Henipavirus Triggering Mechanism

While the attachment and fusion proteins are separate, they still act in concert to translate the receptor binding event into conformational rearrangement in the fusion protein, with the attachment protein passing a triggering signal to the fusion protein. A series of studies has analyzed how factors like glycosylation, the strength of G–F complex formation, and interspecies reciprocity affect the ability of a G–F protein pair to facilitate fusion. These studies are well summarized by Iorio et al. [9] and Bose et al. [10]. Of particular significance was the discovery that in Nipah virus, the attachment protein appears to have an initial conformation that occludes the stalk domain, which is then revealed after receptor binding, passing on a triggering signal to the fusion protein [11]. While these discoveries have greatly advanced understanding of triggering, there has been no comprehensive structural characterization of this process, largely due to a lack of structures of *Henipavirus* attachment protein tetramers or complexes of G–F proteins. However, recent years have seen progress in these areas, which will be explored in this review.

### 1.3. New Henipavirus Species

Since the discovery of Hendra and Nipah viruses in 1994 and 1998, respectively [12,13], the *Henipavirus* genus has been characterized as a group of viruses carried by fruit bats (flying foxes, *Pteropus* genus). Nipah was further categorized into both the Malaysia and Bangladesh strains [14,15], and two additional viruses were then categorized as new species, known as Cedar virus [16] and Ghana virus [17] (also referred to as Ghanian Bat virus or Kumasi virus). However, many novel *Henipavirus* species have been discovered in recent years, with most of the newly discovered species existing in rodent reservoirs rather than in bats. Mojiang virus was discovered infecting rats in China in 2012 [18], followed by the discovery of Langya [19], Gamak, and Daeryong viruses [20] all infecting shrews within either China or Korea. In addition, the geographic distribution of Henipaviruses has grown substantially, with new species discovered in Africa, such as Angevokely virus in Madagascar [21] and Melian virus in Guinea [22]. Other new species have even been found as far as Europe and South America, with Denwin virus in Belgium [22] and Peixe-Boi virus in Brazil [23].

**Figure 2 viruses-16-00195-f002:**
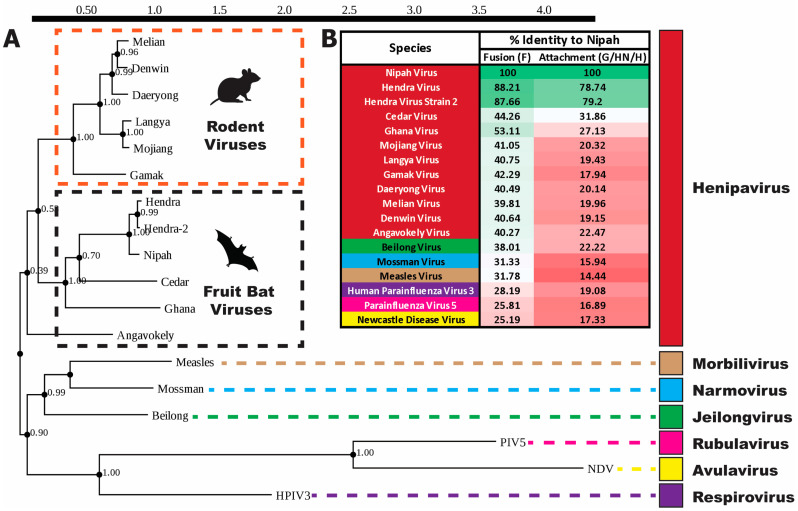
Phylogeny of Henipaviruses. (**A**) Phylogenetic tree of select *Paramyxoviridae* members based on amino acid sequence of viral L proteins, horizontal distances to scale, represent sequence divergence. Scale bar at top in units of substitutions per position. Each node has the support value listed. The fruit bat and rodent-infecting clades of Henipaviruses are highlighted. Measles, Mossman, Beilong, parainfluenza 5 (PIV5), Newcastle disease (NDV), and human parainfluenza 3 (HPIV3) viruses are used as typical examples of the *Morbilivirus*, *Narmovirus*, *Jeilongvirus*, *Rubulavirus*, *Avulavirus*, and *Respirovirus* genera, respectively. Sequence alignment was performed using the Clustal Omega tool [24], and tree building was performed with the Phyml tool using the Clustal Omega tree as the starting point for optimization, the LG substitution model, and tree topology optimization [25]. (**B**) Sequence identity of *Paramyxoviridae* fusion and attachment proteins, relative to Nipah virus. Viruses are colored according to the genus color scheme from A, and identity percentages are colored on a green (greater identity) to red gradient. The following Genbank complete genome entries were used as sources of the L, F, and G/HN/H protein sequences: Hendra, AF017149.3; Hendra-2, MZ229747.1; Nipah, AF212302.2; Cedar, JQ001776.1; Ghana, HQ660129.1; Mojiang, KF278639.1; Langya, OM101130.1; Gamak, MZ574407.1; Daeryong, MZ574409.1; Melian, OK623353.1; Denwin, OK623354.1; Angavokely, ON613535.1; Beilong, DQ100461.1; Mossman, AY286409.1; measles, AB016162.1; HPIV3, EU326526.1; PIV5, OR043657.1; NDV, NC_039223.1 [13,16,18,19,21,22,26,27,28,29,30,31,32].

This proliferation of *Henipavirus* species since the last ICTV profiling of the *Paramyxoviridae* family in 2019 [33] may warrant reorganization of the family’s taxonomy. Currently, the ICTV uses comparison of the amino acid sequence of L, another Paramyxovirus protein that forms part of the RNA-dependent RNA polymerase complex [34], for the classification of Paramyxoviruses. However, many genera are often distinguished by differences in gene expression, such as the lack of expression of a non-structural protein known as C, produced from a reading frame (ORF) that overlaps with the Phosphoprotein in some genera [35] or the expression of additional membrane proteins in some Jeilongviruses and Rubulaviruses [36,37]. Many of the new rodent-infecting Henipaviruses have been found to contain an ORF that encodes a small transmembrane protein overlapping with the F gene [22]. While any new ICTV classification may categorize these newly discovered Henipaviruses as a new genus, partition the Henipaviruses at the subgenus level, or allow clades to be informal designations as with HIV [38], it is clear that a new rodent-infecting group of viruses has emerged (Figure 2), underscoring the high risk of zoonotic transmission in this family. The groupings drawn in Figure 2 are in agreement with a division proposed by Diederich et al. [39], who suggested designation of this group as the *Parahenipavirus* genus.

## 2. Attachment Proteins

### 2.1. Attachment Protein Head Domain

Early characterization of *Henipavirus* attachment proteins revealed critical differences in the host receptors being used as compared with other *Paramyxoviridae* attachment proteins, as well as the lack of hemagglutinating or neuraminidase activities [12,13], distinguishing Henipaviruses from the many Paramyxoviruses, such as the parainfluenza viruses, that were dependent on sialic acid for entry. Structural characterization of the *Henipavirus* attachment proteins revealed, however, that even with this divergence from other Paramyxoviruses, the receptor binding domains retained a highly conserved six-bladed beta propeller architecture similar to the influenza neuraminidase [40] (Figure 3A). Ephrin B2 and B3 (EFNB2, B3), cell surface proteins that are ligands for Eph family tyrosine kinases [41], are the receptors targeted by Nipah and Hendra viruses (NiV/HeV) [42,43]. Structural studies showed the G–H loop of EFNB2/3 from residues 107–125 inserting into the same face of the head domain as where the sialic acid binding site is found in other Paramyxoviruses [44,45,46] (Figure 3A). It was noted that the structure of the NiV/HeV head domain changes only slightly after binding to the EFN receptors, with a subtle shift being noticeable in one of the loops from blade 6 (B6) proximal to the G–H loop insertion site [40,47,48]. More notable was the shift in EFNB2 residue Trp125, which rearranges by over 8Å compared with other EFNB2 structures [40,49,50,51]. In addition to the insertion of the G–H loop into the head domain, salt bridges and hydrogen bonds with other regions of EFN and the receptor binding face of the head domain contributed to the very high affinity interaction between NiV/HeV and their receptors, far higher than other known Paramyxoviruses [40,47]. Further studies revealed the structural basis for differences in receptor binding affinity between Nipah and Hendra viruses, with Nipah being noted for having a higher affinity to EFNB2 [52,53]. In particular, key residues in the binding pocket were noted as being less hydrophobic in Hendra virus as compared with Nipah virus, reducing the hydrophobic character of the pocket into which the EFNB2/B3 G–H loop inserts. However, even with these differences between NiV and HeV, the overall architecture of their head domains remains strikingly similar (Figure 3A).

The sequence identity for *Henipavirus* proteins is low, dropping to below 40% for most species compared with NiV or HeV, which have ~79% identity to each other for their attachment proteins. Although the overall architecture of the head domain is conserved, both sequence variations and local conformational changes ultimately alter the receptor preference. Cedar virus (CedV) is a *Henipavirus* member notable for being nonpathogenic in mammals and for its altered receptor preference [16]. CedV lacks receptor binding capability to EFNB3 but demonstrates EFNB1 binding on a level equivalent to that of B2 binding. Furthermore, CedV is also capable of binding, albeit more weakly, to several A-class Ephrins, specifically EFNA2 and A5 [54]. The structural feature responsible for this shift in receptor preference is a reduction in the number of binding pockets found in the G–H loop binding site that accommodate the insertion of four large hydrophobic residues in EFNB2. In CedV, one of the four pockets is eliminated, and another is enlarged, allowing for easier insertion of the larger residue in EFNB1 (Y121 vs. L121), explaining the inability of NiV and HeV to utilize EFNB1. In addition to these structural changes, there are conformational changes at several loops at the periphery of the head domain in CedV as compared with NiV (Figure 3A). This trend of change in receptor preference was also noted in Ghana virus (GhV), which is still capable of utilizing EFNB2 but not B3 or any of the A-class receptors as CedV is [55]. Conformational changes are still noted at peripheral areas of the head domain, but the primary receptor binding pocket is very similar to NiV/HeV, allowing for a very similar conformation of the bound EFNB2 G–H loop (Figure 3A). While this primary site remains conserved, some of the secondary electrostatic interactions outside of the main site are eliminated in GhV, leading to reduced binding of GhV to EFNB2 compared with NiV.

Among the Henipavirus attachment proteins with known structures, Mojiang virus (MojV) is the most divergent from NiV, being part of the newly forming clade of rodent-infecting Henipaviruses. Although the MojV-G head domain shares the classic Paramyxovirus beta propeller architecture, conformational changes relative to NiV are notable not only in peripheral areas but in the primary binding site as well [56] (Figure 3A). The binding pockets discussed in the previous paragraph are no longer present, eliminating the ability of MojV to bind to EFNB2/B3. In the same study, attempts to identify the receptor for MojV found no binding either to CD150, used by measles virus, or to sialic acid, used by several *Paramyxoviridae* genera. One of the most recently discovered Henipaviruses, Langya virus (LayV) is most closely related to Mojiang virus and is also incapable of binding to EFNB2/B3 [4,19]. This establishes that the rodent-infecting clade of Henipaviruses may be very functionally distinct in terms of both structure and receptor tropism.

The antigenicity of Henipavirus head domains is an important aspect of their immune recognition as the G protein has been found to elicit somewhat higher titers of neutralizing antibodies than the F protein in immunization studies [59]. Several neutralizing antibodies bind directly on the receptor-binding face of the head domain, such as m102.3, an antibody identified by scanning an antibody library using HeV-G proteins. Antibodies have three complementarity determining regions (CDRs) in their structure that are largely responsible for determining their specificity. The third CDR of the heavy chain (CDR-H3) of m102.3 inserts into the receptor binding pockets at the center of the head domain, displaying a similar mode of binding to that of EFNB2/B3 [57] (Figure 3B). Another antibody that binds a similar site is HENV-26, an antibody recovered from an individual exposed to the HeV-G vaccine developed for horses [58]. HENV-26 shares a similar binding mode to m102.3, inserting its CDR-H3 into the receptor binding pocket, although HENV-26 relies on contributions from all CDRs, not just from H2 and H3 as in m102.3 (Figure 3B). From the same study, another antibody known as HENV-32 was structurally characterized, with its binding site revealed to be at the dimerization face of the head domain rather than the receptor binding site (Figure 3B). Several structural studies on Paramyxovirus attachment proteins had detected the oligomerization of head domains [48,60], but the recent structure of the tetrameric NiV-G protein solidifies the biological relevance of head domain dimerization, explaining the neutralization capability of HENV-32 [3]. Of additional interest is the antibody nAH1.3, demonstrated to bind to the opposite face of the head domain, away from the receptor binding site [3] (Figure 3B). The mechanism of neutralization for this antibody is unclear. For these antibodies, the impact of the variability within the genus limits the potential for broad neutralization. HENV-26/32 and nAH1.3 were both noted to lack cross-reactivity with CedV, GhV, and MojV [3,58].

In summary, the head domain of the Henipaviruses utilizes a conserved architecture that can accommodate considerable sequence variation, leading to species-specific receptor recognition. The apparent ability of the Henipaviruses to mutate critical receptor binding residues, even going as far as to completely change receptor preference, highlights the ability of these viruses to diversify, establishing new species that can infect new tissues and hosts, with potentially different pathogenic profiles.

### 2.2. Attachment Protein Tetrameric Structures

While structural information on the head domains of Henipavirus attachment proteins is now widely available, visualization of the quaternary structure of the attachment proteins in their tetrameric state remains sparse. As a result, in order to contextualize the limited available Henipavirus attachment protein structures that extend beyond the head domain, here we will also include in our comparisons attachment protein structures of other Paramyxoviruses, where there is more, albeit still limited, structural information available.

**Figure 4 viruses-16-00195-f004:**
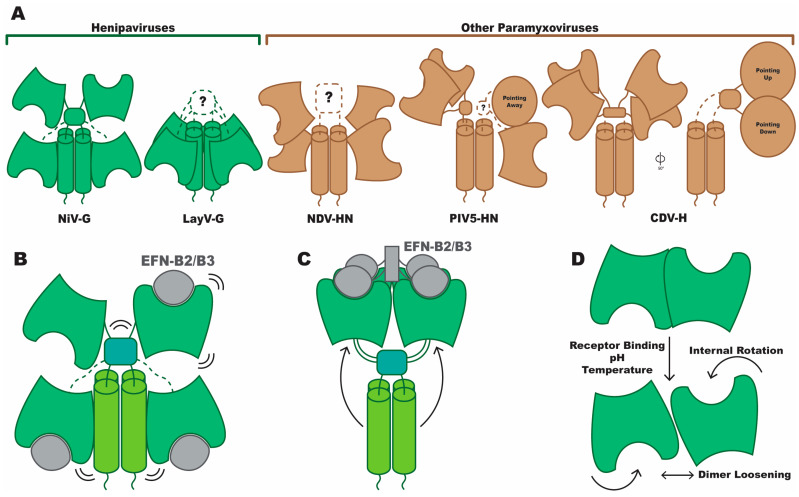
Tetrameric structure of Paramyxovirus attachment proteins. Comparison of Paramyxovirus attachment protein quaternary structures and illustrations of hypothesized conformational dynamics. (**A**) Schematic representations of Paramyxovirus attachment protein tetramers, grouped by those that are Henipaviruses (green) and those from other genera (brown). (**B**) A hypothesized response to monomeric EphrinB2/B3 binding by NiV-G, where the quaternary structure does not significantly change, but the dynamics of the protein, illustrated here as vibration or flexibility, are altered and propagated through the protein. (**C**) A hypothesized response to dimeric or clustered EphrinB2/B3 binding by NiV-G, where the quaternary structure does change, revealing the stalk domain. (**D**) A representation of several combined observations of head domain dimers loosening their interaction or rotating in response to receptor binding or pH or temperature change. PDB IDs: NiV-G, 7TZX, 7TY0 [3]; LayV-G, not released [4]; NDV-HN, 31TE [61]; PIV5-HN, 4JF7 [62]; CDV-H, 7ZNY [63]. NiV: Nipah virus; LayV: Langya virus; NDV: Newcastle disease virus; PIV5: parainfluenza virus 5; CDV: canine distemper virus.

Just as with head domains, it appears that the overall topology of attachment protein ectodomains is conserved in Paramyxoviruses. Proceeding from the N terminus at the cytosolic end of the viral membrane, the attachment protein ectodomain consists of a helical stalk domain that leads into a neck domain, from which a linker region connects to the head domain. However, the orientation of head domains and their positioning relative to the stalk domain vary dramatically from species to species. A cryo-EM structure of the NiV-G ectodomain tetramer was able to define the structure of the stalk domain, each of the four head domains, and the neck domain that connects the two regions, a domain either partially or wholly undefined in most other structures [3]. In this arrangement, two head domains can be seen positioned downward on the stalk, proximal to the viral membrane, with the receptor binding sites facing the viral membrane. The two remaining heads are positioned above the neck, distal to the viral membrane, with the receptor binding site of one of the heads facing toward and the other facing away from the viral membrane. This arrangement is referred to as “two heads up, two heads down” (Figure 4A). A recent preprint details the structure of the LayV-G ectodomain tetramer, currently the only other tetramer structure of a Henipavirus [4]. In contrast to the NiV-G structure, all the head domains are in what could be described as a “down” conformation, with all receptor binding sites pointing toward the viral membrane (Figure 4A). Unlike in the NiV-G structure, the neck domain was not resolved, leaving the connectivity between head and stalk chains uncertain. While it is possible that the structures of Henipavirus G proteins more closely related to NiV, such as HeV and CedV, would show much greater levels of structural similarity, so far it appears that quaternary structures can be quite distinct, even within the genus.

A survey of the structural information that exists outside of the genus can begin with a crystal structure of the Newcastle disease virus (NDV) attachment protein tetramer (called HN for NDV), where the C-terminal end of the stalk domain and the positions of the four head domains were resolved [61]. Each receptor binding site was positioned pointing away from the stalk domain, with one tilted slightly toward and the other slightly away from the viral membrane, on each side of the stalk. No density was observed for the neck domain or linkers to the head domains, meaning that head domains could not be confidently assigned to the appropriate stalk domain chain (Figure 4A). A structure of the parainfluenza virus 5 (PIV5) HN tetramer revealed an arrangement where one dimer within the tetramer was similar to the NDV HN positions, but the other two heads were positioned significantly more distal to the viral membrane [62]. Contrasting the head domain positions with the NDV structure, this configuration was identified as “two heads up, two heads down,” but asymmetrically so, unlike NiV-G (Figure 4A). While this structure was the first for PIV5 to define both the head and stalk domains, previous crystal structures of both of the stalk in isolation and the ectodomain where only the ligand-bound heads were resolved suggested a different arrangement [46,64]. Combining the isolated stalk structure with the dimer of dimers arrangement seen in the head domains suggested a “four heads up” arrangement. As this head arrangement was ligand bound, it suggested the possibility that the two configurations may represent pre- and post-receptor bound states. Also adding to the list of available structures is a cryo-EM structure of the canine distemper virus (CDV) H protein [63]. This arrangement finds the head domains positioned somewhat similarly to the NDV structure, with a pair on each side of the stalk, one receptor binding site pointed toward and the other away from the viral membrane. However, the striking difference is in the offset of these head domains from the stalk domain. From a side view, it is seen that all head domains are to the side of the stalk, rather than in the same plane like in most of the other structures (Figure 4A). This structure also defines a portion of the neck domain, being the only structure outside of the NiV structure to define contributions from each protomer to the neck domain, even if the connection to the remainder of the stalk is not resolved.

These diverse quaternary structures begin to establish a set of conformations that can be accessed by Paramyxovirus attachment proteins. What remains unclear is the extent to which any differences in conformation are driven by sequence variation across the divergent genera or instead are all relevant conformational states that may be sampled by these proteins as they bind to receptors and facilitate fusion. With the limited structural data, especially with only one structure available for each species, it is unclear how consistent conformation will be across different experimental setups, including factors such as concentration, purification conditions, or crystallization conditions. Furthermore, most structural studies of attachment ectodomains utilize various exogenous domains that assist in stabilization of the tetramer. A tomographic study on clinical virions of human parainfluenza 3 virus, a non-*Henipavirus* Paramyxovirus, suggested organization of the attachment protein as a dimer [5], perhaps owing to this distinction. However, the low resolution of the tomographic data prevents confirmation of this finding. Regardless, more structural data are needed to address this question, for both Henipaviruses specifically and the *Paramyxoviridae* family as a whole.

### 2.3. Conformational Response to Receptor Binding

Structural studies have yet to reveal the full landscape of attachment protein quaternary structure or any large-scale conformational changes brought about by receptor binding; however, alterations to dynamics and functionality have been revealed by biophysical assays, including second harmonic generation (SHG), a spectroscopic means of detecting a change in an SHG-active label relative to the probe [65,66]; total internal reflection fluorescence (TIRF) microscopy [67]; and hydrogen–deuterium exchange mass spectrometry (HDX-MS) [68]. Contextualized with other structural data, these assays suggest that several factors affect the dynamics of attachment proteins, such as the oligomerization state of the receptor and environmental factors like temperature and pH. One study evaluated the binding of Ephrin B2 to the NiV-G ectodomain tetramer and determined that receptor binding alters the conformational dynamics of NiV-G, with differential responses noted between monomeric and dimeric Ephrin B2 [69]. Specifically, SHG signal was altered by monomeric and dimeric receptor binding, indicating a shift in the positioning of an SHG-active dye bound to NiV-G, which could be caused by some combination of the dye adopting a new average orientation or a new distribution of orientations. HDX-MS analysis revealed four sites that had increased levels of exposure after monomeric EFNB2 binding relative to the unbound state. These included two sites in the stalk domain and two loops in the head domain opposite the receptor binding site that either interact with the stalk domain when the head domains are in the “down” position or are involved in head dimerization when in the “up” position. However, complementary binding assays using an antibody previously demonstrated to have increased affinity for the receptor-bound state, mAb45 [70], found no substantial change in antibody binding after NiV-G binding to monomeric EFNB2. This was confirmed with negative stain electron microscopy (NSEM) analyses of monomer-bound and unbound NiV-G, which did not reveal any notable conformation shift after receptor binding, finding that in both cases, NiV-G adopted the two heads up, two heads down conformation where the stalk is occluded. In this study, an Fc tag was used to enable the dimerization of EFNB2, but this approach allowed for the dimeric receptor to crosslink NiV-G tetramers, leading to aggregated particles, so NSEM could not compare dimer-bound with unbound NiV-G. However, binding to dimeric EFNB2 led to increased mAb45 binding, as expected. This result established that the oligomeric state of the receptor plays a role in the effects on NiV-G conformation and dynamics. This finding was later supported using full-length G and F proteins, assessing both fusion competency and protein mobility and colocalization through TIRF [71]. With the G and F expressed on a cell in contact with a supported lipid bilayer (SLB), His-tagged EFNB2 was attached to either NTA groups in the lipids, allowing for mobility through the lipid bilayer, or NTA-PEG groups directly attached to the surface, ensuring individual EFNB2 molecules would be immobile. Only when EFNB2 was mobile was a conformation change in F detected. In summary, the oligomerization of the receptor influences the functionality of G, at least in NiV, and likely more broadly in the Henipaviruses, allowing conformational changes in G that are postulated to trigger fusion. While monomeric EFN binding may not be sufficient to induce these dramatic changes, it may still alter protein dynamics in a way that can be transduced through the structure, potentially playing a role in the triggering process (Figure 4B).

To supplement this analysis of attachment protein dynamics, we again turn to structural information outside the *Henipavirus* genus. A recent tomographic study of the human parainfluenza virus 3 (hPIV3) attachment–fusion complex (HN-F) [5] has revealed a different conformation for the dimeric head domain pair from the previously determined crystal structure of the hPIV3 head domain dimer [72]. Consequently, to properly fit the head domain model to the tomography map, it was necessary to fit each head individually, ultimately resulting in a dimeric interface that was rotated and more loosely packed as compared with the crystal structure dimer. The authors highlighted similar shifts in the interaction of head domain dimers across several Paramyxoviruses, including in two crystal structures of the Newcastle disease virus HN head domain dimer, one at physiological and one at low pH [45].

Taken together, when applied to the quaternary structure of *Henipavirus* attachment proteins, these studies suggest three significant steps of how receptor binding initiates conformational changes. First, monomeric receptor binding does not cause large-scale quaternary rearrangements but may still alter dynamics such as flexibility (Figure 4B). Second, binding to a dimerized receptor leads to the opening of the quaternary structure, revealing the previously occluded stalk domain (Figure 4C). For these first two points, there is likely to be little applicability to non-*Henipavirus* Paramyxoviruses as the receptors for other genera, such as sialic acid or CD150, are unlikely to dimerize in a similar manner. Finally, shifts in the interaction between head domains may be part of the conformational cascade that leads to fusion activation, either as a precursor to receptor binding, being driven by factors such as pH change, or as a direct response to receptor binding (Figure 4D).

## 3. Fusion Proteins

### 3.1. Antigenicity of Fusion Proteins

Whereas the head domain is the dominant antigenic region of the attachment protein, neutralizing antibodies targeting the fusion protein can be found in a greater diversity of its domains, and many NiV and HeV neutralizing antibodies have been characterized (Figure 5A). Domain III (DIII) forms the apex of the pre-fusion conformation and contains regions critical to the functionality of the protein, namely, the fusion peptide and one of the heptad repeats (HRA). Thus, it is not surprising to find neutralizing antibodies binding to these epitopes, which likely function by preventing these regions from rearranging and completing their fusion-promotion roles. DIII-targeted antibodies that have been structurally characterized include mAb66 [73], 12B2 [74], 4H3, and 2D3 [75] (Figure 5B). Although the Henipavirus fusion proteins are not heavily glycosylated, several key glycosylation sites affect the distribution of neutralizing epitopes. One glycan present in NiV-F near the end of the DIII helix before the N-terminal beginning of the fusion peptide seems to prevent antibody binding in this site [75], with the antibodies instead binding at sites further toward the apex of the protein (Figure 5B). The apex glycan in NiV-F has a somewhat greater involvement in antibody binding, being at the periphery of the epitope for both mAb66 and 12B2. Both studies investigated the role of the apex glycan in antibody binding by evaluating glycan deletion mutants. In the case of mAb66, deletion did not significantly affect binding, whereas for 12B2, binding was notably higher with the glycan-deleted mutant.

The side face of the prefusion trimer is also a common antibody binding site. There are several structures of antibodies bound to this region, revealing epitopes that often span both DI and DIII. Antibodies 1F5 [74] and 5B3 [78] both bind HRA, neutralizing likely through a similar mechanism to that of the more apex-proximal DIII antibodies. Antibodies 1H8 and 1A9 [75] both bind on the side face of the trimer at quaternary epitopes across multiple domains, with their epitopes shifted more toward the viral membrane relative to 1F5 and 5B3 (Figure 5B). A small portion of the 1A9 epitope includes interactions with several of the surface-exposed residues in the fusion peptide, associating it with one of the few highly conserved sequences in *Henipavirus* fusion proteins. Several antibodies bind to an epitope mainly composed of DII residues. Antibody 2B12 binds across DI and DII in a recess on the side face near to HRB, and 1H1 binds almost exclusively to DII with a small portion of the epitope found on the neighboring protomer’s DIII [75] (Figure 5B). Due to the conformation of DII remaining relatively unchanged through conversion to the post-fusion state, 1H1 can bind both to pre- and post-fusion trimers. In general, both the heavy and light chains contribute heavily to the binding of all antibodies noted here. Unlike antibodies that rely on long CDR loops, such as the receptor binding site-targeted antibodies described previously that mimic the receptor G–H loop or those that must navigate dense clusters of glycans [79], a variety of heavy and light chain arrangements seem to be viable for targeting the fusion protein.

While these antibodies have generally been found to be cross-neutralizing for NiV and HeV, none have yet been found to bind to any of the more divergent Henipavirus species, including the newly discovered clade of rodent Henipaviruses. An examination of how these epitopes differ in some of these more recently discovered species reveals the basis for the lack of cross-reactivity. Considering the sequences of Langya, Mojiang, and Daeryong viruses, for all sites described above, there are many semi- or non-conservative amino acid substitutions relative to the Nipah virus sequence (Figure 5C). The 1H1 epitope in DII and one of the DIII helices that forms a part of the 4H3 epitope are particularly variable. Among these sites, the side face spanning DI–DIII may be the least variable, offering some possibility that this may be a promising site for the elicitation of broadly neutralizing antibodies, yet most of the antibodies binding in this site are seen to bind large, cross-domain epitopes, leaving them vulnerable to sequence variation. While the structural variability between the rodent Henipaviruses and NiV/HeV is not as dramatic as their sequence variability, the differences that are notable are all in important antigenic sites. Recent structural studies of Langya and Mojiang virus F proteins revealed large shifts compared with those of NiV/HeV-F in the HRA β-Hairpin [6], in a loop on the side face near the DI–DIII boundary, and in the long DIII helix that precedes the cleavage site [77] (Figure 5D). While the fusion protein structure offers a wide array of neutralizing sites for immune response, the high level of divergence within the genus makes broad neutralization a significant challenge and indicates that new strains that evade previous immunity may emerge with relative ease.

### 3.2. Fusion Triggering

**Figure 6 viruses-16-00195-f006:**
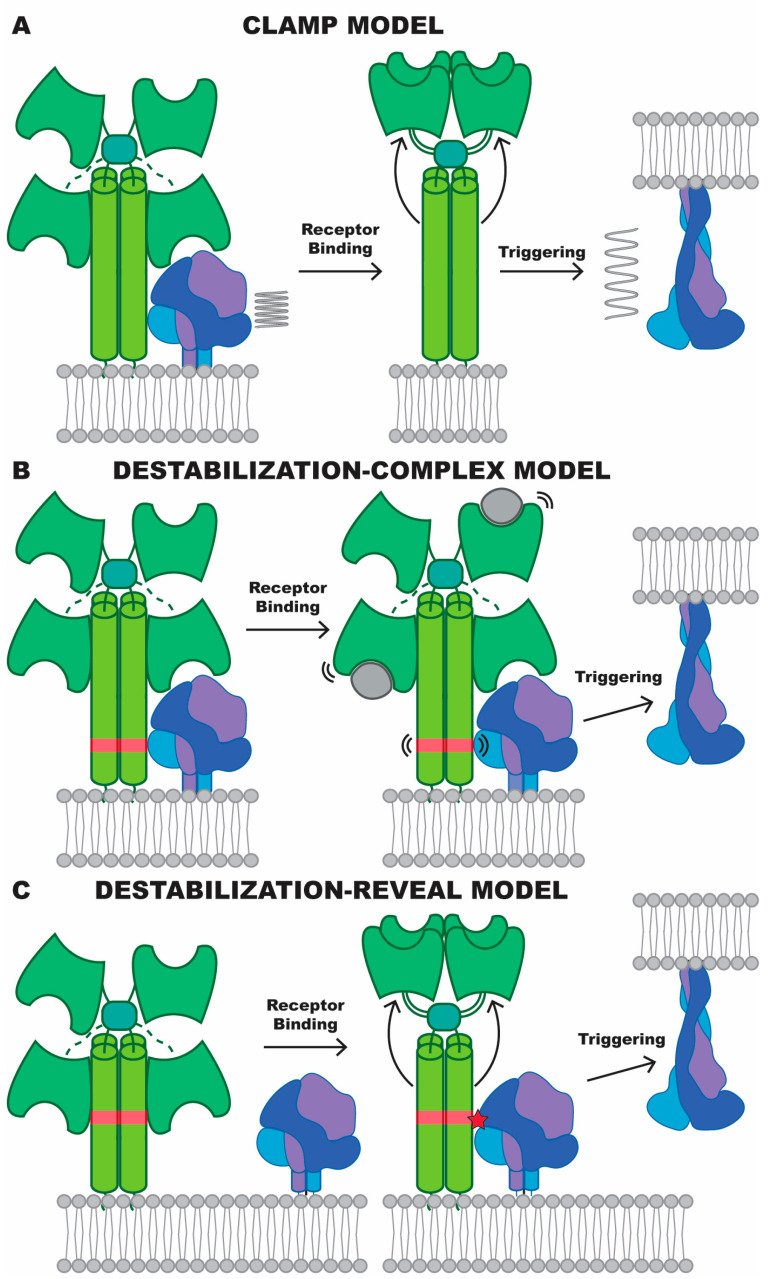
Fusion triggering in Henipaviruses. Illustrations of several hypothesized models of *Henipavirus* fusion triggering. To reduce the complexity of the figure, the transmembrane and cytosolic domains are not illustrated. (**A**) The clamp model, depicting an unstable pre-fusion state that is restricted from rearrangement by interactions with the attachment protein. Conformational rearrangement in the attachment protein, thought to occur through receptor binding, would allow spontaneous conversion of the fusion protein. (**B**,**C**) The destabilization model, where the fusion protein, metastable without assistance, is actively converted to the post-fusion state by interaction with the attachment protein. (**B**) The destabilization–complex model, where the attachment and fusion protein form a complex before triggering. A triggering region of the stalk (red) may even be in contact with the fusion protein but is inert until receptor binding. (**C**) The destabilization–reveal model, where the two proteins do not form a complex before triggering. Receptor binding causes large conformational changes that reveal previously hidden regions of the stalk domain, which can then interact with and destabilize the fusion protein.

The most critical role of the fusion protein is its ability to be triggered by the attachment protein to undergo conformational rearrangement, a process that is yet to be structurally defined. Previous information has established models for *Paramyxoviridae* fusion triggering, known as the association and dissociation models [9]. In both cases, receptor binding to the attachment protein is thought to begin the triggering process, with the association model proposing that an altered attachment protein conformation allows for an interaction with the fusion protein and transference of the triggering signal and the dissociation model proposing that an attachment–fusion protein complex is dissociated by receptor binding, leading to triggering. The basis for these models is discussed in depth by Iorio et al., who identified Henipaviruses as conforming to the dissociation model mechanism of triggering. At the time, it was believed that the attachment protein acted as a clamp on an unstable fusion protein, where release of the fusion protein during the triggering event would allow spontaneous conformational conversion from its pre-fusion to post-fusion form (Figure 6A). However, as first noted by Bose et al. [10], the clamp model is inconsistent with the noted ability of fusion proteins to exist in their pre-fusion conformation without the interaction of the attachment protein. Bose et al. specifically referenced a series of cell-based, biophysical, and antigenic studies that demonstrated this [80,81,82,83,84], and since then, several studies have obtained high-resolution structures of *Henipavirus* fusion proteins in their pre-fusion conformation without using stabilizing mutations or other factors that limit conformational conversion [4,6]. Although these studies revealed populations of both pre- and post-fusion F within the same samples, the presence of significant amounts of pre-fusion F without any extraneous stabilizing factor suggests that an attachment protein head domain clamp cannot be the only factor preventing the F protein from transitioning to its post-fusion form. Furthermore, the ability of headless attachment protein mutants to trigger fusion would suggest that a particular region of the stalk carries the triggering signal, likely inconsistent with a mechanism where the attachment protein solely has the role of restricting fusion protein movement by acting as a clamp [11,83,85]. Taken together, current evidence suggests *Henipavirus* fusion triggering proceeds through a method closer to the association model, where the attachment protein acts on the fusion protein, destabilizing the pre-fusion conformation.

There are several possible pathways through which this destabilization could occur. Central to a model describing fusion triggering through attachment protein-mediated destabilization is identification of how the fusion protein maintains metastability. A recent structure determination of the Langya virus fusion protein, in a state of pre- to post-fusion conversion, has revealed that conformational control of the fusion peptide helps maintain the pre-fusion conformation [6]. Both the fusion peptide and the surrounding hydrophobic pocket are well conserved, consistent with an important evolutionary function for this arrangement. Disruption of this pocket may allow the fusion peptide to leave its pre-fusion arrangement and irreversibly convert conformation. Previous studies have suggested that this pocket is part of a domain that likely interacts with the attachment protein [86,87], proposing a plausible mechanism for how the attachment protein may perform this destabilizing role. This means of fusion triggering would be compatible with both the existence of a G–F complex prior to receptor binding or G–F interactions only occurring after receptor binding. To distinguish this model from the previous association model, it can be referred to generally as the destabilization model. The first possibility under this model is that G–F complexes exist prior to receptor engagement, possibly with the fusion protein and stalk domain interacting (Figure 6B). The fusion-triggering regions of the stalk are in an inert state and only perform their triggering role after receptor binding, where the dynamics of the protein are altered such that this stalk region can now act to destabilize the fusion protein. Under this model, the ability of headless stalk domains to trigger fusion [11,83,85] could be explained by the dynamics of the headless stalk mimicking that of a receptor-bound protein. The other possibility is that attachment and fusion proteins do not form a complex prior to receptor binding. While previous experiments have been able to co-immunoprecipitate the G–F complex from Nipah virus [70,88], another study was unable to detect any colocalization of NiV-G and -F moving through a lipid bilayer [71]. In this case, the occluded stalk domain carries the triggering signal but is inaccessible to the fusion protein until after receptor binding, which moves any “down” head domains away from the stalk (Figure 6C). Given the early indication that the quaternary arrangement of head domains differs greatly across the genus, it is possible that no single mechanism for conformational conversion of the fusion protein is followed by all Henipaviruses.

## 4. Summary

Henipaviruses are emerging pathogens that have demonstrated pandemic potential. The risk to public health is underscored by the current lack of approved vaccines or therapies for *Henipavirus* infection in humans. Structural biology has frequently been the foundation of vaccine and drug development, and this review has focused on the current state of structural biology knowledge for Henipaviruses, specifically in the areas of antigenicity, conformational dynamics, and mechanism of fusion promotion. The concerted action of the attachment and fusion proteins to facilitate fusion is a major mechanistic departure from what is known of other class I viral fusion proteins, creating a unique challenge in understanding the fusion process. The observed conformational diversity of the attachment proteins and identification of sources of metastability for fusion proteins begins to answer this challenge, but a deeper understanding will be required before efforts to stabilize these proteins or target critical elements of the fusion process will be broadly successful. The significant antigenic diversity of these proteins establishes an additional challenge to vaccine development, which is often successful by targeting conserved epitopes that are critical to functionality. Henipaviruses have shown a remarkable ability to conserve their architecture even with high sequence variability, severely limiting options for developing broadly potent vaccines or therapies. However, with recent advances in structural biology, namely, improvements in the quality, range of sample preparation types, and throughput of electron microscopy, the tools necessary to address these challenges are now available.

## Figures and Tables

**Figure 1 viruses-16-00195-f001:**
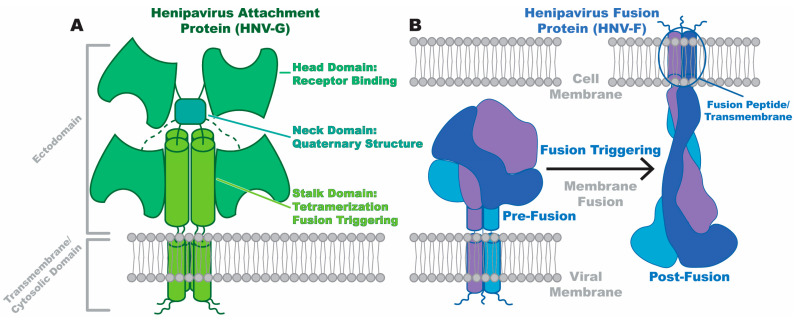
Henipavirus glycoproteins. (**A**) Schematic representation of a *Henipavirus* attachment protein, based on the tetrameric Nipah virus G ectodomain structure [3]. The head, neck, stalk, transmembrane, and cytosolic domains are labeled and colored in different shades of green. (**B**) Schematic representation of the *Henipavirus* trimeric fusion protein, based on structures of the Langya virus F ectodomain structure in pre- and-post-fusion conformation [6]. Individual protomers of the trimer are colored in different shades of blue. The viral and cell membranes, which become one membrane after fusion, are also labeled.

**Figure 3 viruses-16-00195-f003:**
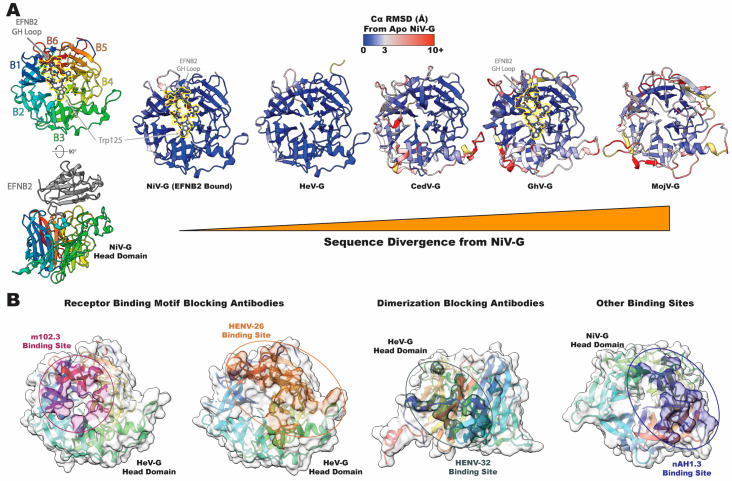
Henipavirus head domain structure and antigenicity. Structural diversity of the *Henipavirus* head domain and comparison of NiV/HeV antigenicity. (**A**) Cartoon representation of head domains from diverse Henipaviruses. NiV-G is shown on the left, colored from the N to the C terminus in rainbow coloration, noting each of the six blades of the beta propellor architecture. The receptor, EFNB2, is shown in gray and from the top view as sticks colored by heteroatom and highlighted with a yellow outline. Head domains from increasingly sequence diverse Henipaviruses are arranged left to right, with residues colored by their C⍺ divergence (Å) from NiV-G. (**B**) Footprints of neutralizing antibodies that target the head domain. Antibodies are grouped by the region of the head domain they target. PDB IDs: NiV-G (EFNB2), 2VSM [40]; HeV-G, 6PD4 [53]; CedV-G, 6P72 [54]; GhV-G, 4UF7 [55]; MojV-G, 5NOP [56]; HeV-G (m102.3), 6CMG [57]; HeV-G (HENV-26), 6VY5; HeV-G (HENV-32), 6VY4 [58]; NiV-G (nAH1.3), 7TY0 [3]. NiV: Nipah virus; HeV: Hendra virus; CedV: Cedar virus; GhV: Ghana virus; MojV: Mojiang virus.

**Figure 5 viruses-16-00195-f005:**
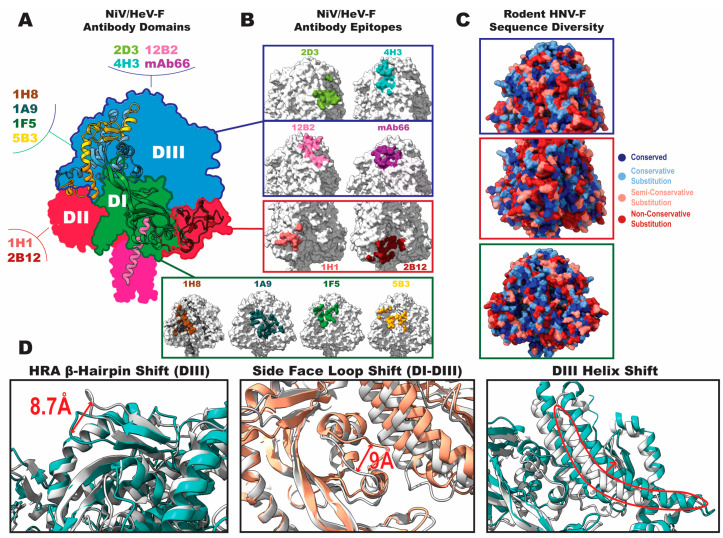
Antigenicity of Henipavirus fusion proteins. Illustration of the footprints of neutralizing antibodies known to bind NiV/HeV-F and examination of how these sites vary across the *Henipavirus* genus. (**A**) Cartoon structure of one protomer of the Henipavirus trimer in pre-fusion conformation, colored by region, and overlaid on a silhouette of the full trimer, colored by domain (DI, green; DII, red; DIII, blue; fusion peptide, orange; HRA, yellow; HRB, pink). Antibodies are associated with the domain where they are known to bind. The DI antibodies all make some contact with DIII. PDB ID: 5EVM [76]. (**B**) Footprints showing each antibody’s binding site, colored uniquely for each antibody and grouped by domain. (**C**) Surface view of LayV-F [6], colored by sequence conservation across the following species: Nipah, Langya, Mojiang, and Daeryong. Colored by residue conservation according to Clustal Omega output [24]. (**D**) Structural divergence between NiV-F [76] (white), from PDB: 5EVM; LayV-F (teal), from PDB: 8FEJ (left) [6] and 8FMX (right) [77]; and MojV-F (brown), from PDB: 8FMY [77].

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
