# Peer review of "Structural Studies of Henipavirus Glycoproteins"

_viruses, 2024, doi:10.3390/v16020195_

Round 1

Reviewer 1 Report

Comments and Suggestions for Authors

Major comments-

·       Section 1.3 on species is a standalone segment that is perhaps best placed in the very beginning before the discussion of surface glycoproteins (section 1.1). It seems especially odd sandwiched between Triggering mechanisms and Attachment proteins.

·       This is a very well written review. A table or figure with sequence comparison and identities for both the attachment and fusion glycoproteins from all bat and rodent viruses discussed (perhaps highlighting some key residues involved in receptor binding for some viruses), would greatly add to the discussion and understanding. Please consider including this.

Line 155-156: what’s the sequence identity between NiV and HeV attachment proteins?

·       Line 207, “…bind to the opposite site of the head domain..”: Is this the “opposite” site or a site that is adjacent or away from the receptor binding site?

·       Figure 5D: Yellow highlights and distances are barely visible, please consider revising.

Other comments-

·       Section 4, Discussion: This is best replaced with ‘Conclusions’ or ‘Summary’ since the review itself is an extended discussion of a research topic.

·       Line 31, “tetrameric dimer of dimers”: this is a confusing. Did you mean, ‘…tetramer, or a dimer of dimers’?

·       Line 55-57: the three shades of blue for fusion protein are hard to distinguish. It may be helpful to label the domains for the fusion protein as well. Line 372: fusion peptide is hard to see, does this have two distinct segments? Line 123-124: the receptor shown in sticks is very hard to see.

Line 83 and 216-220: please include references. 

Comments on the Quality of English Language

Minor typos may need to be corrected.

Author Response

Reviewer 1:

We thank the reviewer for their very helpful comments which have helped improve the visibility of the figures and avoid confusion in several key parts of the text.

Major Comments:

Section 1.3 on species is a standalone segment that is perhaps best placed in the very beginning before the discussion of surface glycoproteins (section 1.1). It seems especially odd sandwiched between Triggering mechanisms and Attachment proteins.

We had previously considered moving this section to the beginning, as it is a departure from the structural focus. However, we ultimately felt that to lead with this section does not as clearly establish the theme of the review. As the review sections focus on the structural diversity that is now more highlighted across the newly discovered species, we felt this section would provide a logical ending to the introduction. It would be our preference to maintain this section’s current positioning but will defer to the editor’s preference.

This is a very well written review. A table or figure with sequence comparison and identities for both the attachment and fusion glycoproteins from all bat and rodent viruses discussed (perhaps highlighting some key residues involved in receptor binding for some viruses), would greatly add to the discussion and understanding. Please consider including this.

A table with sequence identities relative to Nipah Virus for F and G proteins has been added to Figure 2, and the caption has been updated accordingly.

Line 155-156: what’s the sequence identity between NiV and HeV attachment proteins?

The sequence identity between NiV and HeV attachment proteins is 79%. This information has been added to the sentence.

Line 207, “…bind to the opposite site of the head domain.”: Is this the “opposite” site or a site that is adjacent or away from the receptor binding site?

This sentence has been rephrased to clarify that it is not on the receptor binding site, and instead on the ‘opposite face.’

Figure 5D: Yellow highlights and distances are barely visible, please consider revising.

The yellow highlights have been changed to red, with larger lines and text.

Other Comments:

Section 4, Discussion: This is best replaced with ‘Conclusions’ or ‘Summary’ since the review itself is an extended discussion of a research topic.

The section title has been changed to “Summary.”

Line 31, “tetrameric dimer of dimers”: this is a confusing. Did you mean, ‘…tetramer, or a dimer of dimers’?

To clarify that the protein is a tetramer and not any higher order multimer, the sentence now reads: “disulfide bonding allows for formation of a tetramer (dimer of dimers).”

Line 55-57: the three shades of blue for fusion protein are hard to distinguish. It may be helpful to label the domains for the fusion protein as well. Line 372: fusion peptide is hard to see, does this have two distinct segments? Line 123-124: the receptor shown in sticks is very hard to see.

The shades of blue have been changed for the fusion protein in figure 1 and in figure 6 to match. The GH loop in figure 3 is now highlighted yellow for better visibility.

Line 83 and 216-220: please include references. 

The sentence at line 83 includes the references for Angevokely and Melian viruses. Regarding line 216, we have deleted the sentence as it was confusing. The sentence implied that the diversity of receptors used by Henipaviruses is unique, and supporting this statement would require a detailed comparison of viral families that we believe would be outside the scope of this review. The final sentence of the paragraph still stands on its own in highlighting the ability of the species to diversify.

Reviewer 2 Report

Comments and Suggestions for Authors

This manuscript has summarized the structural studies of Henipavirus glycoproteins. This is a well-written and comprehensive review paper can be an invaluable resource in staying updated and gaining deeper insights into the entry mechanism of Henipavirus. I recommend that this paper be accepted.

Author Response

Reviewer 2:

This manuscript has summarized the structural studies of Henipavirus glycoproteins. This is a well-written and comprehensive review paper can be an invaluable resource in staying updated and gaining deeper insights into the entry mechanism of Henipavirus. I recommend that this paper be accepted.

We thank the reviewer for their positive assessment of our paper.

Reviewer 3 Report

Comments and Suggestions for Authors

The article is an interesting review for a narrow circle of specialists about the structure of surface proteins of henipaviruses responsible for fusion with the cell. There are several minor flaws in the article, which are listed below.

I would also like to say that I understand the authors who had read and summarized many literary sources to write the review. And they would like to give all this information to readers. However, the abundance of additional information complicates the understanding of this highly specialized material. I would suggest that authors review their text again and try to simplify the presentation of information where possible. This will allow a wider audience to understand what has been written and will increase the citation rate of the article.

1.The location of the fusion protein relative to the viral and cellular membrane is not clear in Figure 1B. Where in the picture is which one? The captions in Figure 1A are green, those in adjacent 1B and then black. It is logical to lead to monotony.

2.Figure 3 comes before it is mentioned in the text. It's better to move the picture after linking to it.

3.Lines 137-139 mention the diversity of the G-H loop in different viruses and provide clarification about certain amino acid residues. It would be more clear to provide the amino acid aligment of this region between the same viruses, which are represented by three-dimensional structures in Figure 3.

4.The term CDR-H3 appears in line 195; for non-specialists, it is necessary to explain what it is and expand the abbreviation.

5.Lines 190-211 contain a fairly detailed description of a number of antibodies and the places of their influence. The article is already complex, perhaps this part can be simplified or shortened so as not to distract the reader.

6. In Figure 4 (as well as in 3), it would be clearer to the reader if the abbreviated names of viruses were expanded in the figure description.

7. Line 251. The structure with one receptor biding site pointing up and the other down seems very unusual for me. It is not clear from the text how reliably such a structure is permitted. Is it possible to provide a ref. to the publication after this sentence?

8. Line 308 and 310. Ephrin B2 appears as EphrinB2, befor it was EPNB2. You need to either use the old abbreviation or put a space befor B2.

9. Line 298. Paragraph 2.3. As I already mentioned, the review is quite complex and factually rich. Mentioning additional methods in this section complicates the overall picture. To make it easier to understand, it would be good to shorten this section to include the final facts about сonformational response to receptor binding and provide links so that the reader can refer to the methods used if desired.

10. In Figure 6, as in Figure 1, it is not clear why protein F turns upside down.

Author Response

Reviewer 3:

The article is an interesting review for a narrow circle of specialists about the structure of surface proteins of henipaviruses responsible for fusion with the cell. There are several minor flaws in the article, which are listed below.

I would also like to say that I understand the authors who had read and summarized many literary sources to write the review. And they would like to give all this information to readers. However, the abundance of additional information complicates the understanding of this highly specialized material. I would suggest that authors review their text again and try to simplify the presentation of information where possible. This will allow a wider audience to understand what has been written and will increase the citation rate of the article.

We thank the reviewer for their helpful comments that allowed us to improve the clarity of some of the figures and define unfamiliar terms. While we agree that discussion of some of the biophysical studies of Henipavirus proteins may be a departure from the structural focus of the review, their findings have major implications on proposed models for the fusion activation process, which we feel is an important contribution of this review.

1.The location of the fusion protein relative to the viral and cellular membrane is not clear in Figure 1B. Where in the picture is which one? The captions in Figure 1A are green, those in adjacent 1B and then black. It is logical to lead to monotony.

Labels have been added for viral and cell membrane in 1B. The cell membrane is also shown above the prefusion state as well. The captions in 1B have been changed to be blue, matching 1A.

2.Figure 3 comes before it is mentioned in the text. It's better to move the picture after linking to it.

Figure 3 has been moved to after the first paragraph of the section.

3.Lines 137-139 mention the diversity of the G-H loop in different viruses and provide clarification about certain amino acid residues. It would be more clear to provide the amino acid alignment of this region between the same viruses, which are represented by three-dimensional structures in Figure 3.

The G-H loop is a part of the human EFNB2/B3 protein and is not variable. The difference in position noted was for one residue that adopts a different conformation when bound by virus attachment protein rather than the human Eph protein that binds EFNs.

4.The term CDR-H3 appears in line 195; for non-specialists, it is necessary to explain what it is and expand the abbreviation.

A sentence has been added to introduce the complementarity determining regions (CDRs).

  1. Lines 190-211 contain a fairly detailed description of a number of antibodies and the places of their influence. The article is already complex, perhaps this part can be simplified or shortened so as not to distract the reader.

Antigenicity is an increasingly relevant topic as the field works towards effective and broadly neutralizing vaccines. Therefore, the detailed description of these neutralizing sites is important to catalogue for comparison with results of future vaccination studies.

  1. In Figure 4 (as well as in 3), it would be clearer to the reader if the abbreviated names of viruses were expanded in the figure description.

Expanded names of viruses have been added in figure 3 and 4 legends.

  1. Line 251. The structure with one receptor biding site pointing up and the other down seems very unusual for me. It is not clear from the text how reliably such a structure is permitted. Is it possible to provide a ref. to the publication after this sentence?

The publication is referenced in the preceding sentence, line 251 is further discussing the reference. The cryo-EM structure is at a high enough resolution to confirm this conformational arrangement.

See:

Wang, Zhaoqian, Moushimi Amaya, Amin Addetia, Ha V. Dang, Gabriella Reggiano, Lianying Yan, Andrew C. Hickey, Frank DiMaio, Christopher C. Broder, and David Veesler. "Architecture and Antigenicity of the Nipah Virus Attachment Glycoprotein." Science 375, no. 6587 (2022): 1373-78.

  1. Line 308 and 310. Ephrin B2 appears as EphrinB2, before it was EPNB2. You need to either use the old abbreviation or put a space before B2.

A space has been added to the two instances of EphrinB2.

  1. Line 298. Paragraph 2.3. As I already mentioned, the review is quite complex and factually rich. Mentioning additional methods in this section complicates the overall picture. To make it easier to understand, it would be good to shorten this section to include the final facts about conformational response to receptor binding and provide links so that the reader can refer to the methods used if desired.

In this section, we are synthesizing findings from these biophysical studies to propose modifications to the fusion triggering models that are not directly made by any of the studies. Because of this, we feel it is important to describe these methods and a short summary of the experiments. In particular, we expect that second harmonic generation will not be familiar to the typical reader.

  1. In Figure 6, as in Figure 1, it is not clear why protein F turns upside down.

As part of responses to other comments, we have added labelling that clarifies where the cell and viral membranes are. The F protein turns upside-down to indicate that it has now fused with the target cell membrane, positioned opposite the original orientation. Figure 1 now includes additional membranes and labels that try to clarify this change in direction as well as a clarification in the caption.

Reviewer 4 Report

Comments and Suggestions for Authors

The manuscript by May et al. thoroughly examines the structural biology of henipaviruses, specifically focusing on the glycoproteins crucial for viral entry: the attachment (G) protein and the fusion (F) protein. Belonging to the family of Paragraphmyxoviridae, henipaviruses represent a significant threat to both human and animal health. The elucidation of the structural dynamics of these viral glycoproteins is vital in the development of targeted vaccines and therapeutics. This review offers an extensive guide to the structural biology of henipavirus glycoproteins and mentions other biophysical experiments that support the mentioned models. In conclusion, the review highlights the gaps in understanding the henipavirus entry mechanism, including recent structural studies on novel henipaviruses and the authors' recent work.

The review is nicely written, and I do not have many comments regarding the review itself. Comments listed here might improve the review and make it more accessible to the readers:

Specific Comments:

·         Paragraph 21: The introduction could better provide more background on the virus family properties and genomic architecture to set the stage for readers unfamiliar with the Paragraphmyxoviridae classification.

·         Paragraph 27: Specify the H protein is a type II membrane glycoprotein.

·         Paragraphs 27-35: Address the confusion from the absence of a broader explanation of Paragraphmyxoviruses. The cited tomography study uses hPIV3 (human Paragraphinfluenza virus 3), which is not a henipavirus; it needs clarification.

·         Paragraph 44: Detail the mechanism of the fusion peptide anchoring in the host membrane. The use of fusion peptide is quite ambiguous.

·         Figure 1 Comments:

o   The paper mentions structural studies of Henipavirus glycoproteins but omits the single-particle structure of Nipah G. It would be beneficial to present the actual atomic model as a reference for the presented schematic.

o   Modify the current schematic to accurately represent the transmembrane region's interaction with the lipid bilayer, including both the transmembrane region and cytoplasmic tail. For the post-fusion F protein, include the fusion peptide anchored in the host membrane. Clearly label the viral and host cell membranes.

o   I would suggest incorporating the actual atomic model obtained by the authors alongside the schematic for a comparative perspective.

·         Figure 2 Comment: Including representative species from other genera within the Paragraphmyxoviridae family, such as Avulaviruses, Respiroviruses, and Rubulaviruses, would provide a more comprehensive overview, beneficial for readers.

·         Paragraph 108: Since Rubulaviruses are referenced, include them in the phylogenetic tree for completeness.

·         Figure 3 Comments:

o   Add Protein Data Bank IDs to the structures either on the figure or in the legend. Include sequence divergence percentages for clarity (min/max).

o   In the accompanying text, discussing electrostatic surface changes on the G protein head and presenting electrostatic potential maps would visually elucidate the differences among species.

·         Figure 4 Comments:

o   Mention PDB IDs if the schematics are derived from structural data.

o   The schematic in part D requires a more coherent representation for clarity. Currently, it is difficult to understand from the schematic what it represents.

·         Paragraph 252: Address the inconsistency in Figure 4A's representation of the "2-heads up, 2-heads down" configuration. The figure looks like 1 up only, so please add some extra text for clarity.

·         Paragraph 291-297: It is important to note that most structural studies of attachment proteins of Paragraphmyxoviruses use a tetramerization domain that might influence the obtained oligomeric state. It would be helpful to mention it in this section as the authors comment on the recent cryo-ET structure in its native state.

·         Figure 5A: Include the missing PDB ID for the structure presented.

·         Figure 5C: Ensure consistency in the presentation of species across structural data; the use of LayV-F glycoprotein here, as opposed to NiV in Figure 3, could lead to confusion. I would suggest using the same reference structure throughout the review for clarity.

·         Figure 5D: I propose replacing one of the LayV-F structures with a HeV structure for a meaningful comparison unless significant differences are noted between the two LayV-F structures. In that case, that should be noted in related publications describing LayV-F structures.

·         Figure 6: Add missing elements, such as the transmembrane and cytoplasmic parts, and include appropriate labels.

Author Response

Reviewer 4:

The manuscript by May et al. thoroughly examines the structural biology of henipaviruses, specifically focusing on the glycoproteins crucial for viral entry: the attachment (G) protein and the fusion (F) protein. Belonging to the family of Paramyxoviridae, henipaviruses represent a significant threat to both human and animal health. The elucidation of the structural dynamics of these viral glycoproteins is vital in the development of targeted vaccines and therapeutics. This review offers an extensive guide to the structural biology of Henipavirus glycoproteins and mentions other biophysical experiments that support the mentioned models. In conclusion, the review highlights the gaps in understanding the Henipavirus entry mechanism, including recent structural studies on novel henipaviruses and the authors' recent work.

The review is nicely written, and I do not have many comments regarding the review itself. Comments listed here might improve the review and make it more accessible to the readers:

We thank the reviewer for their extensive suggestions that have helped us improve the figures as well as clarify parts of the text, especially the use of “up/down” language to refer to head domain positioning and orientation and the observation about the use of tetramerization domains in many of the structural studies.

Specific Comments:

Paragraph 21: The introduction could better provide more background on the virus family properties and genomic architecture to set the stage for readers unfamiliar with the Paramyxoviridae classification.

The terms negative-sense, non-segmented, and RNA have been added to this sentence to add this background.

Paragraph 27: Specify the H protein is a type II membrane glycoprotein.

This wording has been added to the sentence.

Paragraphs 27-35: Address the confusion from the absence of a broader explanation of Paramyxoviruses. The cited tomography study uses hPIV3 (human Parainfluenza virus 3), which is not a Henipavirus; it needs clarification.

Language has been added in that sentence to clarify that this study is from a related, but non-Henipavirus species. Later sections of the paper establish the necessity of looking outside the genus for structural information.

 Paragraph 44: Detail the mechanism of the fusion peptide anchoring in the host membrane. The use of fusion peptide is quite ambiguous.

An additional sentence has been added to explain the role of the fusion peptide.

Figure 1 Comments:

The paper mentions structural studies of Henipavirus glycoproteins but omits the single-particle structure of Nipah G. It would be beneficial to present the actual atomic model as a reference for the presented schematic.

Modify the current schematic to accurately represent the transmembrane region's interaction with the lipid bilayer, including both the transmembrane region and cytoplasmic tail. For the post-fusion F protein, include the fusion peptide anchored in the host membrane. Clearly label the viral and host cell membranes.

I would suggest incorporating the actual atomic model obtained by the authors alongside the schematic for a comparative perspective.

The images have been modified to show where the transmembrane and cytosolic domains are. Text now signifies that the fusion peptide and transmembrane domain are together in the post-fusion state. We feel that addition of the atomic model would make the figure too dense, and the many available structures for G and F proteins are cited and can be investigated further by readers.

Figure 2 Comment: Including representative species from other genera within the Paramyxoviridae family, such as Avulaviruses, Respiroviruses, and Rubulaviruses, would provide a more comprehensive overview, beneficial for readers.

Paragraph 108: Since Rubulaviruses are referenced, include them in the phylogenetic tree for completeness.

The phylogenetic tree figure now includes the additional genera of the Paramyxoviridae family mentioned.

Figure 3 Comments:

Add Protein Data Bank IDs to the structures either on the figure or in the legend. Include sequence divergence percentages for clarity (min/max).

Added PDB IDs to the legend. Sequence identity for F and G proteins has been added to figure 2.

In the accompanying text, discussing electrostatic surface changes on the G protein head and presenting electrostatic potential maps would visually elucidate the differences among species.

There are several factors that might make comparison of electrostatic potential maps misleading. The models used are truncated at differing residues, therefore what might appear to be an electrostatic shift for one species could just be the presence of a residue not modeled in another. Additionally, the modelling of side chains at the peripheral loops of each model could be subject to differences in resolution across the studies. In summary, displaying electrostatic surfaces would require careful evaluation of each structure and detailed assessment of the atomic model, which we believe is outside the scope of this review article.

Figure 4 Comments:

Mention PDB IDs if the schematics are derived from structural data.

Added PDB IDs to figure 4 legend.

The schematic in part D requires a more coherent representation for clarity. Currently, it is difficult to understand from the schematic what it represents.

Text now clarifies what is meant by the arrows in panel D.

Paragraph 252: Address the inconsistency in Figure 4A's representation of the "2-heads up, 2-heads down" configuration. The figure looks like 1 up only, so please add some extra text for clarity.

Up/Down refers to the head domain position relative to the stalk and neck domain, not orientation of the binding site. Language has been changed to use “up/down” to refer to head domain position, and “away/towards/facing” are used to describe the directionality of the receptor binding domain.

Paragraph 291-297: It is important to note that most structural studies of attachment proteins of Paramyxoviruses use a tetramerization domain that might influence the obtained oligomeric state. It would be helpful to mention it in this section as the authors comment on the recent cryo-ET structure in its native state.

We have added to the end of the paragraph noting this point.

Figure 5A: Include the missing PDB ID for the structure presented.

Added to figure legend.

Figure 5C: Ensure consistency in the presentation of species across structural data; the use of LayV-F glycoprotein here, as opposed to NiV in Figure 3, could lead to confusion. I would suggest using the same reference structure throughout the review for clarity.

We have removed the arrows pointing away from panels A and B to prevent confusion that suggests all images are based on the NiV-F model. New text above panel C also helps establish this.

Figure 5D: I propose replacing one of the LayV-F structures with a HeV structure for a meaningful comparison unless significant differences are noted between the two LayV-F structures. In that case, that should be noted in related publications describing LayV-F structures.

Previous structural studies referenced by this review have overlayed NiV and HeV-F structures, and there is virtually a complete overlap. Text has been added that conveys that LayV-F and MojV-F are being compared to the essentially interchangeable NiV/HeV architecture. Additionally, there are indeed differences noted between the LayV-F structures, with the HRA beta hairpin shift being noted in one of the LayV-F structures. This is reflected in the in-line citations, with each citation being linked to the study that demonstrated the structural shift.

Figure 6: Add missing elements, such as the transmembrane and cytoplasmic parts, and include appropriate labels.

While we agree that the missing domains should be added to figure 1 for clarity, they are not being discussed in this figure and we feel adding them would be distracting and cause the figure to become too visually dense. A note was added in the legend that these domains are omitted for simplicity.

Round 2

Reviewer 3 Report

Comments and Suggestions for Authors

The authors have corrected everything that was necessary